# Scaling Up and Distilling Down:
# Language-Guided Robot Skill Acquisition

Huy Ha[1], Pete Florence[2], and Shuran Song[1]

[1]*Columbia University*
[2]*Google DeepMind*

**Abstract:** We present a framework for robot skill acquisition, which 1) efficiently scale up data generation of language-labelled robot data and 2) effectively distills this data down into a robust multi-task language-conditioned visuo-motor policy. For (1), we use a large language model (LLM) to guide high-level planning, and sampling-based robot planners (*e.g.* motion or grasp samplers) for generating diverse and rich manipulation trajectories. To robustify this data-collection process, the LLM also infers a code-snippet for the success condition of each task, simultaneously enabling the data-collection process to detect failure and retry as well as the automatic labeling of trajectories with success/failure. For (2), we extend the diffusion policy single-task behavior-cloning approach to multi-task settings with language conditioning. Finally, we propose a new multi-task benchmark with 18 tasks across five domains to test long-horizon behavior, common-sense reasoning, tool-use, and intuitive physics. We find that our distilled policy successfully learned the robust retrying behavior in its data collection procedure, while improving absolute success rates by $33.2\%$ on average across five domains. All code, data, and qualitative policy results are available at our project website.

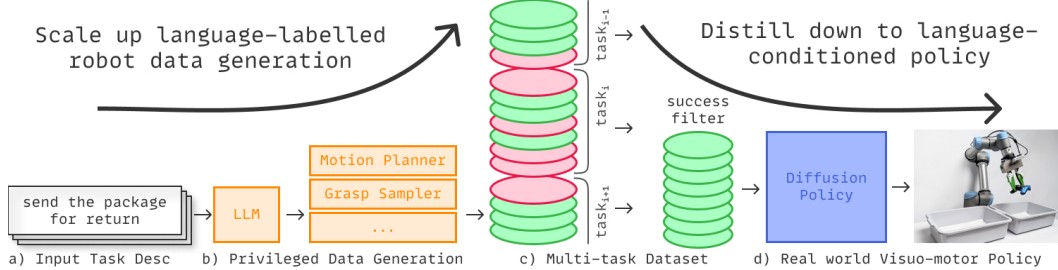

Figure 1: **Language-guided Skill Acquisition** enables scalable robot learning. In the data generation stage, a LLM takes as input task descriptions (a) and uses sampling-based robotic planners and privileged simulation information (b) to perform task-directed exploration. This enables the scaling up of language and task-success labeled dataset generation (c). In the second stage, the dataset is filtered for success and distilled down into a closed-loop language-conditioned visuomotor policy for real world deployment (d).

## 1   Introduction

How can we scalably acquire robust, reusable, real-world manipulation skills? This question has been the driving force behind extensive research in robot learning. Attempts in the field have focused on two primary aspects: First, how to **scale up the data collection** for a diverse range of manipulation skills, which involves efforts such as improving the hardware [1, 2] and software [3, 4] which support demonstration collection, utilization of non-robotics datasets [5, 6], or trial-and-error explorations [7]. The second aspect of this question concerns **effective learning** from the collected data, which delves into exploring effective action representations [8–10] and policy formulations [11, 12] that can robustly model the training data and generalize to novel scenarios.

This paper proposes a new framework that provides a comprehensive solution for both aspects by leveraging language guidance, while using no expert demonstrations or reward specification/engineering. We contribute two key components with our framework:

- **Scaling Up Language-Guided Data Generation:**  Our data-collection policy is a large language model (LLM) which has access to a suite of 6DoF exploration primitives (*i.e.*, sampling-based robot planners and utilities). Given an input task description, this policy first **simplifies** the task by recursively decomposing it into subtasks, resulting in a hierarchical plan (*i.e.*, task tree). Next, this plan is **grounded** into a sequence

7th Conference on Robot Learning (CoRL 2023), Atlanta, USA.

of 6DoF exploration primitives, which generates diverse robot trajectories for the task. Finally, the data collection policy **verifies** the trajectories' success with an inferred success function and **retries** the task until it succeeds. This verify & retry step not only improves the data-collection policy's success, but also adds robot experience on how to recover from failure, an important trait for downstream policy distillation. This data generation approach is scalable, enabling significantly more efficient autonomous task-directed exploration than unguided alternatives (*i.e.*, reinforcement learning) while not being limited by the lack of low-level understanding of the LLM-only solution.

- **Distilling Down to Language-Conditioned Visuomotor Policy:** We distill these robot experiences into a visuo-linguo-motor policy that infers control sequences from visual observations and a natural language task description. To enable effective learning of high entropy, diverse robot trajectories, we extend the diffusion policy [12] to handle language-based conditioning for multi-task learning. This allows the learned policy to be reused and recomposed through language-based planners. We found that our distilled policy successfully learned the robust retrying behavior from its data collection policy, while improving upon its absolute success rate across five domains by 33.2%. Further, we demonstrate that our policy directly transfers to the real-world without fine-tuning using domain randomization.

Our framework combines these two components to get the best of both worlds – leverage LLM's common-sense reasoning abilities for efficient exploration while learning robust and re-usable 6DoF skills for real-world deployment. In summary, the key contribution of this paper is a new framework for visuo-linguo-motor policy learning that is enabled by three novel components:

- A new language-guided data collection framework that combines language-based task planner with 6DoF robot utilities (*e.g.* motion planning, grasp sampling).

- New formulation of diffusion-based policy that effectively learns multi-task language-conditioned closed-loop control policies.

- In addition to our algorithmic contributions, we also contribute a new multi-task benchmark that includes 18 tasks across five domains, requiring long-horizon ($\approx 800$ control cycles), common sense, tool-use, and intuitive physics understanding – capabilities lacking in existing manipulation benchmarks.

## 2 Related Works

**Scaling visuo-linguo-motor data.** In learning vision-and-language-conditioned motor policies for real-world deployment [9, 10, 13–18], one of the most important questions is how to scale up "robot-complete data" – data that has robot sensory inputs (*e.g.* vision), action labels (*e.g.* target end-effector & gripper commands), and task labels (*e.g.* language description, success). The most prevalent paradigm is to use humans to annotate both actions (*e.g.* teleoperation) and language [9, 10, 13–18]. When providing action labels, humans can either provide task-specific [9, 10, 15, 18], or task-agnostic ("play") data [13, 14, 16, 19]. A primary limitation, however, is that data scalability is human-limited.

Other prior works have proposed strategies to enable more-autonomously-scalable data. To scale language annotation, prior works study using visual-language models [20, 21], or procedurally post-hoc provided in simulation [19]. To scale action labels, methods study how to use *autonomous sub-optimal policies* from random [7] to learned [22] policies. Human egocentric videos [6, 23, 24] has also been shown to be relevant to robot learning [5, 25], but *is not robot-complete* (lacks action labels), and requires cross-embodiment transfer. Towards unsupervised exploration, prior works have also investigated evolving environments [26, 27] and embodiments [28], automatic task generation [29], leveraging language guidance [30, 31] and world-model error [32], but have not been demonstrated to scale to 6 DoF robotic skill learning. While these approaches reduce human efforts, they are still limited in optimality, generality, and/or completeness of robot data labels.

Another option for the autonomous data collection policy is to use a model-based policy, *e.g.* task and motion planning (TAMP) [33]. Our approach extends such methods in terms of flexibility and task generality by leveraging LLM's common-sense knowledge. However, in contrast to recent works which use LLMs as the *final* policy [34–40], we use the LLM-based planner as a suboptimal *data-collection* policy. We then distill only successful trajectories into an observable-information [41–43] policy, allowing the distilled policy to improve upon its LLM data collection policy's performance.

**Policy Representations and Multi-task Policy Distillation.** One primary question in visuo-motor learning [44] has been how to represent the policy for effective learning, i.e. to enable high precision, multi-modal robot behavior [2, 11, 12, 45, 46]. Another related question has been how to best train multi-task policies [47, 48], including those conditioned on language [9, 10, 13, 15, 16, 18]. Our work presents the novel formulation of bringing diffusion-based [49, 50] policies [12] into the language-conditioned [51, 52] visuomotor domain. Additionally, prior works in multi-task language-conditioning typically focus on cloning policies from experts, meanwhile we study distilling data from a success-filtered suboptimal policy. Success-filtering [11, 53] can be viewed as the simplest form of offline RL [54].

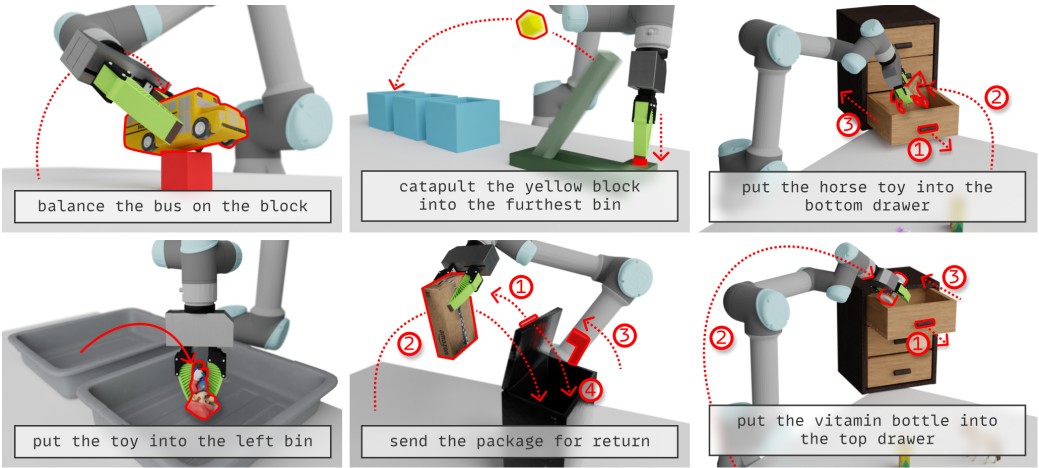

Figure 2: **Benchmark.** We validate our approach on a new multi-task benchmark addressing challenging long-horizon tasks (*i.e.*, 800 control cycles) requiring language understanding (e.g., put [object] to [top] drawer), common sense knowledge (e.g., send a package for return requires raising the mailbox flag), tool-use (e.g., catapult), and intuitive physics (e.g., balance the bus). The tasks are best viewed on our our project website.

## 3 Approach

We propose a new framework for robot learning that performs automatic data collection and policy learning from only a task description. Our design is grounded on four key observations:

- We recognize the importance of random exploration in reinforcement learning, but aim to not be constrained by its inefficiency for long-horizon, sparse reward tasks.

- We acknowledge the usefulness of LLM's common-sense and zero-shot capabilities, but believe language *is not by itself* the ideal representation for robust, rich, and precise robotic manipulation.

- We are inspired by the effectiveness of robotic planning methods, e.g. TAMP, but wish to be flexible to novel tasks and domains and non-reliant on ground truth state during policy inference.

- We aim to achieve the simplicity and effectiveness of behavior cloning in distilling collected robot experience into a policy for real-world deployment, while side-stepping the requirement for costly human demonstrations or play data collection.

Using no human demonstration or manually specified reward, our framework combines the strengths of these four areas into a unified framework for both efficient task-directed exploration and multi-task visuo-linguo-motor policy learning.

**Method Overview.** In the data generation phase, we use an LLM to recursively decompose (§3.1) tasks into a hierachical plan (*i.e.*, task tree) for exploration and ground the plan into sampling-based robot utilities and motion primitives (§3.2). Next, the LLM infers success-detection functions for each task in the plan (§3.3), providing success-labeling. This autonomous data generation process outputs a replay buffer of task-directed exploration experience, labeled with language descriptions and success labels. In the training phase (§3.4), we filter this data for success according to the LLM inferred success condition and distill it into a multi-task vision-and-language-conditioned diffusion policy [12].

### 3.1 Simplify: Task Planning and Decomposition

Given a task description, the first step is to generate a high-level task plan. To improve the flexibility to work with any tasks and 3D assets, we opted for an LLM-based planner to leverage their common-sense and zero-shot reasoning skills. Unlike classical TAMP planners, our framework does not require domain-specific engineering and transition function design to work with new tasks.

Concretely, our recursive LLM planner takes as input the task description, the simulation state, and outputs a plan in the form of a task tree (Fig. 3a). To do so, the LLM first checks whether the task description involves the robot interacting with multiple or only one object. For instance, "move the package into the mailbox" involves opening the mailbox before picking up the package and putting the mailbox in, and should be considered a multi-object task. Meanwhile, "with the mailbox opened, move the package into the mailbox" should be a single-object task. For the base case of single-object tasks, we prompt the LLM to which object part name to to interact. For the case of multi-object tasks, we prompt the LLM to decompose the task into subtasks, and recurse down each subtask.

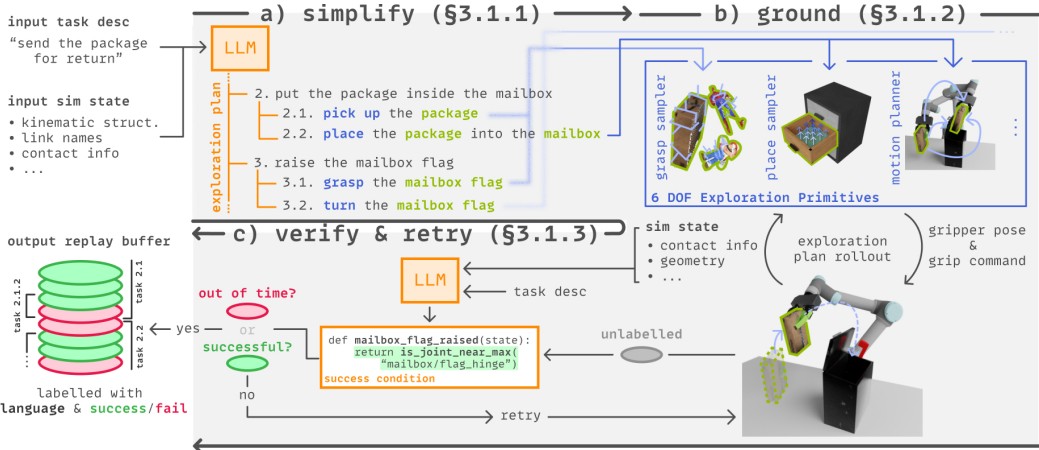

Figure 3: **Language-Driven Robot Data Generation** takes as input the task description and simulation state, and outputs a replay buffer, labelled with language descriptions and success. It starts by using an LLM to simplify tasks recursively (a) until the task involves only one object, resulting in a hierarchical exploration plan. Next, the plan is grounded (b) into a sequence of 6 DOF exploration primitives (*e.g.* grasp samplers, motion planners, etc.) and rolled out in simulation to give an unlabelled robot trajectory. Finally, an LLM infers a success function code-snippet, and uses it to verify (c) and label it with succeeded or failed. If the trajectory failed, the LLM retries the exploration plan with a different random seed (*e.g.* a different grasp pose from the grasp sampler). If the robot succeeds or run out of time, the labeled trajectory is returned.

## 3.2 Ground: Compiling a Plan into Robot Utilities

With the generated task tree §3.1, the next step is to ground the high-level plan into physical actions. Here, the choice of the *low-level robot API* critically defines the system's capability and, therefore, becomes a key differentiating factor between different systems. In principle, there are three desired properties we want to see in the action space design:

- **Flexibility.** Planar actions [10, 37] aren't flexible enough to manipulate prismatic and revolute joints.

- **Scalable.** Namely, actions should not require human demonstrations to acquire [9, 10, 13–16, 35].

- **Language-friendly.** While joint sequences can encode any action, it is not language-friendly.

We propose to ground the LLM's plan with API calls into a set of robot utility functions, which include a sampling-based motion planner, a geometry-based grasp and placement sampler, and motion primitives for articulated manipulation. We refer to these utilities as 6 DOF Exploration Primitives (Fig 3b) because, by virtue of being *pseudo-random*, the sampling-based utilities generate *diverse* robot trajectories, enabling effective exploration for rich 6 DoF manipulation settings. For instance, our grasp and placement samplers samples uniformly amongst all points in the object part's point cloud to find good grasps and placements poses, respectively, which are used as input into a rapidly-exploring random trees [55] motion planner that samples uniformly in joint space. This results in diverse grasps, placements, and motion trajectories connecting grasps and placements.

For each leaf node in the inferred task tree (§ 3.1), the grounding process takes as input the node's task description (*e.g.* "open the mailbox"), its associated object part name (*e.g.* "mailbox lid"), and the simulation state, and outputs a sequence of 6 DoF Exploration Primitive API calls. Using the object part name, we can parse the object's kinematic structure from the simulation state and handle articulated and non-articulated (*i.e.*, rigid, deformable) objects separately. For non-articulated objects, the LLM is prompted to choose the pick & place object names, used to sample grasp and placement pose candidates. For articulated objects (with either revolute or prismatic joints), the leaf node's associated object part name is used to sample a grasp candidate followed by a rotation or translation primitive conditioned on its joint parameters (i.e., joint type, axis, and origin).

**Exploration Plan Rollout.** Each node in the exploration plan is grounded only when it is being executed, where the order of execution follows a pre-order tree traversal. By keeping track of the subtask's state, sub-segments of robot trajectory can be labelled with the subtask's description, thereby providing **dense and automatic text labels** for the trajectory. For instance, all actions taken during the inferred subtask "open the mailbox" can be labeled with both the subtask's description "open the mailbox" and the root task description "move the package into the mailbox".

Since grounding happens only when a task node is visited, each node's grounding process is independent of the other leaf nodes, depending only on the simulation state when it is evaluated. While this simplifies planning significantly, it also means that failed execution can occur. For instance, a grasp candidate may render all placement candidates infeasible.

### 3.3 Verify & Retry: Robustifying the Data Collection Policy

Recall, the planning and grounding step can fail, especially when we consider long-horizon tasks. To address this, we propose a verify & retry (Fig. 3c) scheme, which uses environment feedback to detect failed execution.

**Verify.** For each task, the LLM infers **a success function code snippet** given the task description, simulation state, and API functions to for query simulation state (e.g., checking contact or joint values, etc). This amounts to prompting the LLM to complete a task success function definition that outputs a boolean value, indicating task success. For instance, given the task "raise the mailbox flag", the LLM's inferred code snippet should check whether the mailbox's flag hinge is raised (Fig. 3c, highlighted green).

**Retry.** When a trajectory is labeled failed, the robot retries the same sequence of robot utilities with a different random seed (*i.e.*, for the sampling-based robotic utilities) without resetting the simulation state until the task succeeds. For instance, in the bus balance task (Fig. 2, top left), the robot would repeatedly try different grasp and place candidates until the bus is balanced. In the tree traversal process § 3.2, nodes only yield execution to its parent task when the node's inferred success condition returns true. This design not only leads to higher success rates in data generation but also provides useful demonstrations on **how to recover from failure**. In the output replay buffer, the only failed trajectories are ones which timed-out or led to invalid states (*e.g.* object dropped on the floor).

### 3.4 Language-conditioned Policy Distillation

We extend diffusion policy [12], a state-of-the-art approach for single-task behavior cloning, to the multi-task domain by adding language-conditioning. This policy takes as input a task description CLIP [56] feature, proprioception history, and visual observations, and outputs a sequence of end effector control commands. Following Robomimic [4]'s findings, we use a wrist-mounted view in addition to a global (workspace) view to help with tasks requiring precise manipulation. We use their ResNet18-based [57] vision encoders, one for each view. We found that

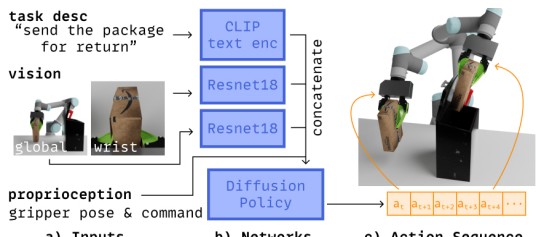

Figure 4: **Language-Conditioned Policy Distillation**. The policy takes as input a task description, two RGB camera views, and gripper proprioception data, and outputs a sequence of gripper poses and closing command.

using only the latest visual observation along with the full observation horizon of proprioception maintains the policy's high performance while reducing training time. When used in conjunction with the DDIM [58] noise scheduler, we found that we could use a $10\times$ shorter diffusion process at inference (5 timesteps at inference, 50 timesteps at training) while retaining a comparable performance. Quantitatively, when using a 10 dimensional action space[*], our policy can be run at $\approx 35Hz$ on an NVIDIA RTX3080.

## 4 Evaluation

Our experiments try to validate two questions: 1) Can our data generation approach efficiently perform task-directed exploration? 2) Can our policy learning approach effectively distill a multi-modal, multi-task dataset into a generalizable and robust visuo-linguo-motor policy?

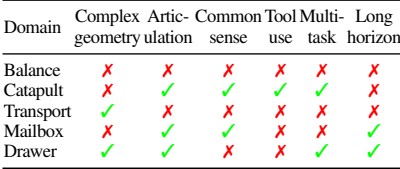

| Domain | Complex geometry | Artic-ulation | Common sense | Tool use | Multi-task | Long horizon |
|---|---|---|---|---|---|---|
| Balance | ✗ | ✗ | ✗ | ✗ | ✗ | ✗ |
| Catapult | ✗ | ✓ | ✓ | ✓ | ✓ | ✗ |
| Transport | ✓ | ✗ | ✗ | ✗ | ✗ | ✗ |
| Mailbox | ✗ | ✓ | ✓ | ✗ | ✗ | ✓ |
| Drawer | ✓ | ✓ | ✗ | ✗ | ✓ | ✓ |

Table 1: **Benchmark Suite.**

**Our Benchmark** contains 18 tasks across 5 domains (Fig. 2 Tab. 1), with the following properties:

- **6DoF & articulated manipulation**, for deadling with complex object geometry and articulation.

- **Geometry Generalization.** In our bin transport domain, the robot must generalize its bin transport skill to unseen object instances, with novel shapes, sizes, and colors.

- **Intuitive physics.** Robots should understand the physical properties of the world and use this knowledge to perform tasks. In the bus balance domain, the robot needs to learn the precise grasping and placement to balance a large bus toy on a small block. In the catapult domain, where the block is placed along a catapult arm determines how far the block will be launched, and, thus, which bin (if any) the block will land in.

- **Common-sense reasoning & Tool-use.** Natural language task description is user-friendly but often under-specifies the task. Common-sense can help to fill in the gaps. In the mailbox domain, given the task "send the package for return", the robot should understand that it not only needs put the package inside, but also raise the mailbox flag to indicate that the package is ready for pickup. In the catapult domain, the robot needs to understand that pressing the catapult's button will activate the catapult, and that the block needs to be placed on the catapult arm to be launched.

---

[*]3 for position, 6 for rotation using the upper rows of the rotation matrix, and a gripper close command

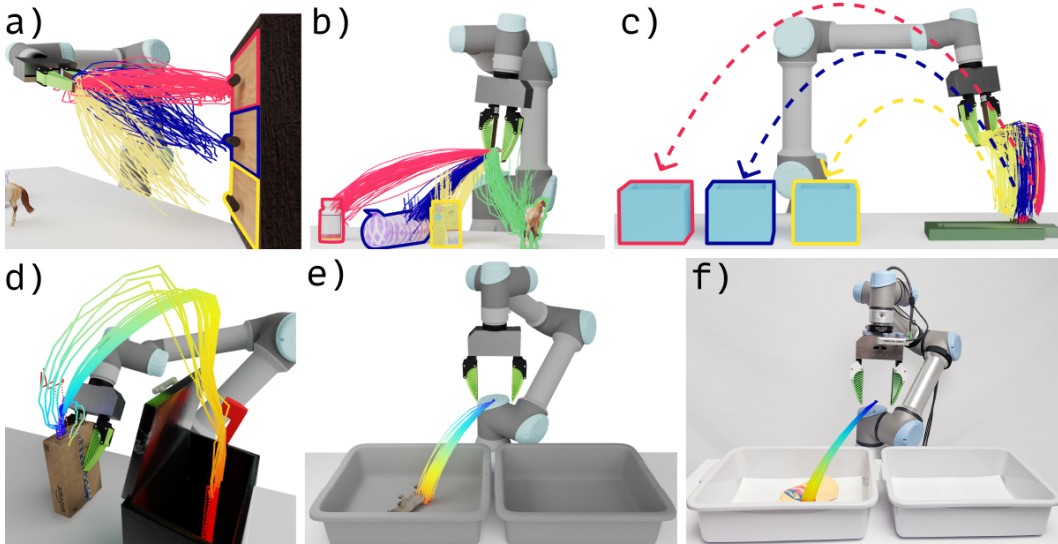

Figure 5: **High Entropy yet Precise Language-Guided Action Sequences.** Running the pseudorandom language-conditioned diffusion process with different seeds on the same observations yields language-consistent (a-c, different colors for different task descriptions), high entropy actions when possible (a-f, object grasping, transports, & placements) and precise actions when necessary (d, narrow mailbox with large package). Further, domain randomization enables a simulation trained policy (e) to generalize to the real world (f).

- **Multi-task conditioning.** Given the same visual observations but different task description, the robot should perform different and task-relevant actions. The catapult domain has 3 tasks for three target bins, and the drawer domain has 12 tasks.

- **Long horizon behaviour.** Our longest horizon domain, mailbox, takes at least 4 subtasks to complete (open the mailbox, put the package in the mailbox while its opened, close the mailbox, then raise the mailbox flag) which can require up to 800 control cycles. In the drawer domain, the robot needs to open the drawer, move the object into the drawer, then close it, which takes about 300 control cycles.

The benchmark is built on top of the MuJoCo [3] simulator, using assets from the Google Scanned dataset [59, 60]. We use a table-top manipulation set-up with a 6DoF robot arm. The task success in evaluation is a manually designed function, instead of LLM generated function used for data collection.

**Metrics.** We report the success rates (%) averaged over 200 episodes in Table 2, a task completion efficiency plot in Fig. 6, and qualitative results in Fig. 5. If a domain has multiple tasks then we report the average performance of all tasks. We also compare different LLMs in Table 4 (10 samples per task) and investigate the sources of error in our system for the mailbox domain in Table 3 (200 trials per execution).

**Data Generation Baselines.** Code-as-Policy [37] is a state-of-the-art approach for using an LLM directly as a robot policy by making state (*e.g.* query present objects) and action primitive API calls to a robot. Given an LLM-inferred code string, they execute the snippet in an open-loop fashion. Crucially, in their table top manipulation setting, they assume access to planar action primitives. Thus, we introduce the following baselines, which build on top of Code-as-Policy and each other as follows:

- **LLM-as-Policy (2D)**: Similar to code-as-policy using planar pick-and-place, but we use ground truth object segmentation instead of their off-the-shelf object detectors [61, 62].

- **(+) 6 DOF robot utils**: Builds on top of the previous baseline by adding access to 6 DOF robot utilities for grasping, placement, motion planning, and articulated manipulation.

- **(+) Verify & Retry**: Adding to the previous baselines, this baseline uses the LLM's predicted success condition to label trajectories and retry failed ones. Since the robot utilities involve pseudo-random samplers (*e.g.* RRT, grasp sampling), retrying the task means running these samplers again using the pseudo-random state and environment state from where failed trajectory left it. Since we use this approach as our data generation policy, it also serves as an ablation of our approach.

**Policy Distillation Ablations.** We compare against BC-Z [15]'s single-task policies which does not use FiLM conditioning (used in their bin emptying and door opening tasks). To understand the effects of our policy learning design decisions in the single-task regime, we fix training time and dataset size (2 days using at least 500 successful trajectories), and provide the following ablations:

- **Action Generation**: Instead of using diffusion processes conditioned on the policy input embedding to decode actions, it is typical use multi-layer perceptrons. Following Jang et al. [15], we use one **MLP** with two hidden layers and ReLU activations for end effector position, one for the orientation, and another for

gripper command. This standard policy architecture is deterministic, and is trained with mean-squared error loss for pose and binary cross entropy loss for gripper command.

- **Action Space**: Besides our absolute end effector pose action space, **Delta-Action** and velocity control spaces is another popular action space choice [4, 15, 63–65]. We also ablate BC-Z's execution action horizon (Exec) while keeping their original prediction horizon (Pred).

- **Observation Encoder**: All approaches encode images using a ResNet18 [57] architecture. Although the original architecture was designed with an average pooling layer, its typical for robotic policies to use a spatial softmax pooling [44] layer instead.

- **Data usage**: **No-Retry** trains on successful trajectories generated from the data generation approach without Verify & Retry, so it does not observe any recovery behavior.

### 4.1 Data Collection Policy Evaluation

**6DoF exploration is critical.** First, we verify different approach's ability to perform and explore in 6DoF, which is crucial for general manipulation. When 6DoF exploration is introduced, we first observe a drop in the average success rate for simple tasks that could be accomplished with planar actions (Balance, Transport, Tab. 2). However, this ability is critical for exploring complex tasks, providing data to improve upon in the

| Approach | Planar | | | 6DoF | | Average |
|---|---|---|---|---|---|---|
| | Balance | Catapult | Transport | Mailbox | Drawer | |
| LLM-as-Policy (2D) | 28.0 | **33.3** | 21.5 | 0.0 | 0.0 | 27.6 |
| (+) 6DoF Robot Utils | 5.5 | 2.5 | 35.0 | 0.0 | 1.3 | 8.8 |
| (+) Verify & Retry | **45.0** | 7.3 | **82.0** | **3.0** | **31.8** | **33.8** |
| Distill No Retry | 67.5 | 38.5 | 32.5 | 0.0 | 22.7 | 32.2 |
| Distill Ours | **79.0** | **58.3** | **80.0** | **62.0** | **55.8** | **67.0** |

Table 2: **Success Rates (%)** for data generation (top) and distillation approaches (bottom) over 200 trials.

later distilling stage. In particular, we observed that 6DoF actions are important for grasping diverse objects with complex geometry (Transport, Tab. 2), and manipulating articulated objects (Drawer, Mailbox, Tab. 2).

Moreover, 6DoF exploration also helps in **diversifying** the data collection strategy, which provides the **possibility to improve upon** in the later distilling stage. For example in the catapult domain, LLM-as-Policy (2D) is only able to solve one of three possible goals (the closest bin) using a deterministic strategy. However, it provides no useful data for learning the other two goals, making it a poor data-collection policy. In contrast, incorporating 6 DOF robot utilities achieves lower but non-zero average success rates in all bins (16.3%, 3.3%, and 2.2%, full table in appendix), which provide much better exploration data for distillation.

| Subtask | Planning | Verify | Execution |
|---|---|---|---|
| Open mailbox | 100 | 100 | 43.5 |
| Put package in mailbox | 100 | 100 | 28.5 |
| Raise mailbox flag | 100 | 100 | 62.0 |
| Close mailbox | 100 | 100 | 94.2 |

Table 3: **Sources & Propagation of Error**. Accuracy (%) of planning, verification, and execution success rate (%) for each mailbox subtask.

**Verify & Retry always helps.** In the verify & retry step, the LLM retries all tasks until they are successful. This simple addition improves performance in all domains, with $2\times$, $3\times$, $8\times$, and $13\times$ in transport, catapult, balance, and drawer domains. Without this crucial step, we observe $0.0\%$ success rate in the mailbox domain, underscoring the difficulty of flawlessly executing long sequences of 6 DOF actions, and the importance of recovery after failure.

**Language Model Scaling.** In addition to the final task success, we provide more detailed analysis of planning and success condition inference accuracy in Tab. 4. We evaluate on the proprietary GPT3 [66] (175B text-davinci-003) and the open LLAMA2 [67] (7B and 13B). We found that Llama models struggles in complex planning domains because they

| Model | Size | Planning | Success |
|---|---|---|---|
| LLAMA2 | 7B | 42.0 | 10.0 |
| | 13B | 62.0 | 48.3 |
| GPT3 | 175B | **82.0** | **91.1** |

Table 4: **LLM Evaluation**.

do not follow instructions provided in the prompts. For instance, in the drawer domain, both models fail to account for drawer opening and closing. However, we observe an upwards trend with respect to Llama model size, with the 13B model outperforming the 7B model by $+20.0\%$ and $+38.3\%$ in planning and success verification accuracy respectively.

### 4.2 Distilled Policy Evaluation

**Robustness In, Robustness Out.** By filtering trajectories with LLM's inferred success condition, distilled policies inherit the robustness of their data collection policies while improving upon success rates ($+23.4\%$ and $+33.2\%$ for no-retry and ours, Tab. 2). Since our distilled policy learned from a robust data collection policy, it also recovers from failures (*e.g.* failed grasps or placements) and continuously retries a task until it succeeds. Meanwhile, since the no-retry distilled policy learned from a data collection policy which did not retry upon failure, it is sensitive and brittle, leading to $-34.8\%$ lower average success rate across all domains compared to ours (Tab. 2).

**High Performance From Diverse Retry Attempts.** Plotting how long policies take to solve the balance task (Fig. 6), we observed that our policy and its data collection policy continuously tries a diverse

set of grasps and placements after each failed attempt until it succeeds. This results in higher success rates as the policy is given more time, and is reflected in their monotonically increasing success rates.

In contrast, baselines plateau after their first grasp/placement attempts. This highlights the synergy of two design decisions. First, the verify & retry step (§ 3.3) is crucial for demonstrating retrying behavior, but is by itself *insufficient* if each retrying action is the identical as the previous one. Instead, opting for a diffusion policy (§ 3.4) for learning from and generating high-entropy, diverse retry attempts (Fig 5) is also essential for high performance.

**Policy Learning Baselines.** We investigate policy learning design decisions on the single-task balance domain, and remove language conditioning. While BC-Z found spatial softmax hurt their performance and opted for a mean pool, we observed using spatial softmax improved performance by +5.0%. Further, we found that switching from delta to absolute action spaces improved success rates +6.5% and +9.5% when using the MLP action decoder and our diffusion action decoder, respectively, confirming Chi et al. [12]'s findings. Lastly, we find that using our pseudo-random diffusion-based action encoder consistently outperforms a deterministic MLP action mappings, regardless of other design decisions.

Figure 6: **Distilled Robustness**. **Our policy** inherits robust recovery from failure behavior from **its data collection policy**, while improving upon success rate.

**Sim2Real Transfer.** We evaluated a policy trained on domain randomized synthetic data in a real world transport task with five novel objects (Fig. 5e). Averaging across ten episodes per object, our policy achieved 76% success rate, demonstrating the effectiveness of our approach in Sim2Real transfer.

| Method | Output | | | | Input | | Success |
|---|---|---|---|---|---|---|---|
| | Generation | Rep. | Exec | Pred | Pool | Proprio | (%) |
| BC-Z | FeedForward | Delta | 1 | 10 | Avg | ✗ | 0.0 |
| | FeedForward | Delta | 4 | 10 | Avg | ✗ | 15.0 |
| | FeedForward | Delta | 8 | 10 | Avg | ✗ | 18.5 |
| Ours | FeedForward | Delta | 8 | 16 | Spatial | ✓ | 29.0 |
| | FeedForward | Abs | 8 | 16 | Spatial | ✓ | 35.5 |
| | Diffusion | Delta | 8 | 16 | Spatial | ✓ | 69.5 |
| | Diffusion | Abs | 8 | 16 | Avg | ✓ | 76.5 |
| | Diffusion | Abs | 8 | 16 | Spatial | ✓ | **79.0** |

## 4.3 Limitations

By using priviledged simulation state information, the LLM can infer success conditions which uses ground truth contact, joint information, and object poses. This means our implementation of the data generation phase is limited to simulation environments, and our policy requires sim2real transfer. Further, Our data generation method relies on existing 3D assets

Table 5: **Policy Learning Ablations**. Action generation using diffusion models [50] robustly outperforms feed-forward models across other policy design decisions.

and environments, which presents a further opportunity for scaling up with assets from 3D generative models or procedural generation. Finally, while our approach's dataset contains text labels and success labels for all subtasks, we have only evaluated its effectiveness in learning the root task. Learning from all subtasks and growing a robot's set of learned, reusable sub-skills over time to enable compositional generalization is left for future work.

## 5  Conclusion

We proposed "Scaling Up and Distilling Down", a framework that combines the strengths of LLMs, sampling-based planners, and policy learning into a single system that automatically generates, labels, and distills diverse robot-complete exploration experience into a multi-task visuo-linguo-motor policy. The distilled policy inherits long-horizon behaviour, rich low-level manipulation skills, and robustness from its data collection policy while improving upon performance beyond its training distribution. We believe that this integrated approach is a step towards putting robotics on the same scaling trend as that of LLM development while not compromising on the rich low-level control.

**Acknowledgments**

We would like to thank Cheng Chi, Zeyi Liu, Samir Yitzhak Gadre, Mengda Xu, Zhenjia Xu, Mandi Zhao and Dominik Bauer for their helpful feedback and fruitful discussions. This work was supported in part by Google Research Award, NSF Award #2143601, and #2132519. We would like to thank Google for the UR5 robot hardware. The views and conclusions contained herein are those of the authors and should not be interpreted as necessarily representing the official policies, either expressed or implied, of the sponsors.

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
