# OpenReview forum: "Scaling Up and Distilling Down: Language-Guided Robot Skill Acquisition"
_robot-learning.org/CoRL/2023/Conference — CoRL 2023 Poster_

### Official Review · Reviewer_t1nk · 2023-06-28

**Confidence:** 5
**Originality:** Very Good
**Technical Quality:** Very Good
**Clarity Of Presentation:** Very Good
**Impact:** 4

**Recommendation:**

Strong Accept: I recommend accepting the paper and will argue for my recommendation even if other reviewers hold a different opinion.

**Review:**

Overall, I enjoyed reading this paper and I think it should be accepted. There are a few issues that I found while reading that I think the authors could help address in the rebuttal.

### Strengths

**************************************Writing and Clarity**************************************: This paper is very well written and clear. The method, at a high level, is also very intuitive and simply (though pretty complicated at a lower-level with respect to the LLM) and the authors did a good job separating details from intuition. Figures are also excellent throughout the paper.

**************************Experiments:************************** Overall, experiments are solid and the baselines are sensible. Results demonstrate the method is necessary under this problem formulation compared to the baselines. The environment is also very diverse in task distribution and demonstrates many axes of comparison.

************************Motivation:************************ Upon reading this paper at first, I was thinking “why not just use the planners if they are already good enough to *************generate data************* by using the LLMs to call the planners when a new task arrives?*************”************* However, the authors resolve this problem by using difficult environments in which even good planners struggle to solve the task. This does lead to the question of “can these failed trajectories be used to improve the planners?” but that is likely less scalable than just training a policy.

### Weaknesses

************************Motivation:************************ This work claims to be very general, but does seem to require quite a bit of domain-specific tuning. Robot pick-and-place is the task that the authors focus on but to make this work generally (e.g., navigation + manipulation for general robots) requires difficult design of the robot utility primitives and likely also difficult design of the “verify + retry” mechanism. Of course, to be fair, the authors don’t claim that this will work on those more general cases so this is somewhat of a nitpick.

Also given how big the LLM prompt is in the supplementary, it likely requires careful tuning of the prompt mechanism along with a very expensive/expressive/large language model, AND extensive scene descriptions to work. Is this approach very scalable to real-world environments? How robust can these success detectors be without perfect object-state information and the like? The authors experiment only in sim.

****************************************Method explanation:**************************************** How do the authors use the failed trajectories? L182: “In the output replay buffer, the only failed trajectories are ones which timed-out or led to invalid states (e.g. object dropped on the floor).” Are these failed trajectories used to train the policy? If so then how does behavior cloning differentiate between successful and failed trajectories? If not, then please clarify this in the text.

************************Experiments:************************

- Baseline explanations: L245 "ground truth segmentation,” segmentation of what exactly?
- How much data is generated for each task? This is included in the supp (500 successful trajs) but should be in the main paper for context.
- It would be beneficial to the academic community, especially among those who want to reproduce this type of work without the ability to work with industry-level language models or paying for GPT-3/3.5/4/4++ credits, to have an ablation study with some other open-source LLMs. VICUNA, LLAMA, ALPACA, and many other animal-inspired open-source LLMs are now flourishing while sometimes claiming to match GPT-3.5/4 performance on certain tasks. Please include some ablation study of this — this is important to the academic community and is the main reason for my “weak” accept.
- 50 seeds per task but no error bars on anything?
- Not asking for this to be included because this is asking too much, but did the authors try VLMs that can caption images? E.g., LLAVA-13B, GPT4, PALM (Bard), or more traditional captioning models? These models could help detect success conditions *********possibly********* more robustly than generating success checkers.
- No robot experiments, but at this point these simulated environments are very similar to real-world robot tabletop manipulation so the application is clear for CoRL.

### Minor Points

- Related work: you can add this recent work to the **************scaling visuo-linguo-motor data************** paragraph for scaling up pre-annotated data: https://arxiv.org/abs/2306.11886
- Related work: please add this work to the same section for work very similar to code-as-policies: https://arxiv.org/abs/2209.11302 (L78)
- L142: function → functions
- L238: an → a

**Quality Of The Limitations Section:**

Limitations are addressed clearly

**Questions For Rebuttal:**

Please see above for questions, I interspersed them throughout the review.

**Robotics Focus:**

Highly relevant to robotics but no hardware experiments

**Summary Of Paper:**

The authors propose a robust framework for using LLMs to call planners to execute long-horizon manipulation tasks to generate data for training a policy.

**Summary Of Recommendation:**

I enjoyed this paper. It is a weak accept mainly due to the lack of ablation studies that would be useful for the research community (LLM ablations) and some questions I have regarding the motivations.

If these are addressed during the rebuttal, I would easily recommend this paper with a strong accept.

---

### Official Review · Reviewer_kQcE · 2023-07-20

**Confidence:** 4
**Originality:** Very Good
**Technical Quality:** Good
**Clarity Of Presentation:** Very Good
**Impact:** 4

**Recommendation:**

Weak Reject: I recommend rejecting the paper, but will not argue for my recommendation if the majority of other reviewers have a different opinion.

**Review:**

Strengths:

- The paper studies how to use LLMs to guide data collection, which is interesting and novel. This work can potentially pave the way for exciting new directions.

- An ablation study is included in Table 2 with detailed discussions, which provides useful insights about how each component contributes to the performance.

- The paper is mostly well-written and easy to follow. The data generation pipeline is clearly explained.

- I like the design of the testing tasks. These tasks are sufficiently complex while grounded in real-world applications.

Weaknesses:

- It is hard to justify the contributions regarding the diffusion policy (distillation party) since no comparisons or ablations are provided in that regard. I would suggest the authors run ablations using policy networks adapted from BC-Z (Jang et al.), Language-Play (Lynch et al.), and other prior work.

- It would be better to include discussions about prior work on unsupervised task generation such as POET (Wang et al.), MAP-Elites (Fontaine and Nikolaidis), ATR(Fang and Migimatsu et al.), PLR (Jiang and Dennis et al.); and language-guided exploration such as ELLA (Mirchandani et al.) and ELLM (Du and Watkins et al.).

- It is unclear how such methods can be applied to the real world.



**Quality Of The Limitations Section:**

Limitations are addressed clearly

**Questions For Rebuttal:**



**Robotics Focus:**

Highly relevant to robotics but no hardware experiments

**Summary Of Paper:**

This paper aims to learn language-conditioned control policies by collecting data in simulation. Using Large Language Models (LLMs) and predefined motion primitives, the proposed method can generate diverse tasks in simulation to train a language-conditioned diffusion policy to solve instruction following tasks. The learned policy is compared with a variant of Code-as-Policies on a set of simulated tasks and achieves better performance.

**Summary Of Recommendation:**

The paper proposes a reasonable method for learning language-conditioned policies by generating data in simulation. My main concerns are the lack of justification of the contributions regarding the diffusion policy part and insufficient discussions about prior work. If these concerns are addressed, I will consider raising my rating.

---

> ### Author Response · Authors · 2023-08-14
> **Rebuttal Follow-up**
>
> We would like to thank you again for your detailed review and the suggested related works and policy distillation baselines. During the rebuttal, we have:
>  - Included policy distillation results with BC-Z, and systematically varied many policy learning design decisions (Table below, identical to Table 5 in the revised PDF). The results showed that diffusion-based action generation was a key design decision to achieving better performance.
>  - Added all of your suggested related works in L83-86.
>  - Demonstrated our system in the real world. We observe that its robust retrying behavior (best viewed on our [anonymous website](https://scalingup-distillingdown.github.io/)) was important for high performance.
>
> We hope we have addressed all your concerns. Please don't hesitate to ask for more details!

---

> ### Author Response · Authors · 2023-08-15
> **Last Day of Discussion Phase**
>
> Dear reviewer kQcE,
>
> Today is the last day of discussion period, we would like to check in to see if you have any additional question we could help answer. Thank you!

---

> ### Comment · Reviewer_kQcE · 2023-08-15
> **Response to Rebuttal**
>
> Dear authors,
>
> I appreciate your efforts on the ablative study and the real-world demo. I have a few questions regarding these additional experiments.
>
> - In your results, the diffusion policy significantly outperforms the feed forward baseline that use the same observation and action spaces. The performance improvements seem to be more dramatic than the results in most of the tasks reported in prior work on diffusion policies such as [Chi et al.](https://arxiv.org/pdf/2303.04137.pdf) (see Table I, II, IV in their paper). Expressive models such as the diffusion policy could be usually helpful when the action distribution is multi-modal, which does not seem to be the case for the testing tasks designed in this paper. Could the authors provide further explanation about the performance improvement introduced by their method? Is it caused by the diffusion policy in a similar way to prior work or specific design choices introduced in this paper?
>
> - In the real-world experiments, note that the policy is only trained on 22 objects in simulation and the authors demonstrate that the policy can generalize the 5 unseen real-world objects. The data augmentation seems to be applied to the lighting and the background instead of the objects. Could the authors provide explanations about how the object-level generalization is achieved?

---

> > ### Author Response · Authors · 2023-08-15
> > **Synergy between Verify & Retry and Multi-modal actions from Sampling-based Robot Utilities**
> >
> > > Expressive models such as the diffusion policy could be usually helpful when the action distribution is multi-modal, which does not seem to be the case for the testing tasks designed in this paper.
> >
> > Our choice of using diffusion policy is precisely motivated by the multi-modal action distribution.
> > In our case, our sampling-based grasping and motion planner output very multi-modal actions, for example switching which "side" of an object to grasp, or which part of configuration space to use in order to plan collision-free motions. The entropy of the data is best visualized on our anonymous [supplementary website](https://scalingup-distillingdown.github.io/), which shows our distilled policy using both [top-down and side-grasps](https://scalingup-distillingdown.github.io/videos/compressed_traces/balance/2cf23f/output/bus.mp4) to solve the task.
> >
> > > Could the authors provide further explanation about the performance improvement introduced by their method?  Is it caused by the diffusion policy in a similar way to prior work or specific design choices introduced in this paper?
> >
> > We hypothesize that the gap between feed-forward and diffusion action generation is larger in our experiments due to higher entropy/ **greater multi-modality** in our generated data. While human demonstrations could contain multiple common human-like grasp modes, our “grasp and placement samplers samples uniformly amongst all points in the object part’s point cloud” (L147-148 from the original submission), which generates all possible successful grasp modes.
> >
> > > Could the authors provide explanations about how the object-level generalization is achieved?
> >
> > We hypothesize that our distilled policy learns a very general strategy, which can be applied to any object: keep trying diverse grasps until the object is moved to the target bin. This strategy is best viewed on our website ([controller](https://scalingup-distillingdown.github.io/videos/realworld/controller.mp4), [toy plush](https://scalingup-distillingdown.github.io/videos/realworld/monster.mp4), [rubik’s cube](https://scalingup-distillingdown.github.io/videos/realworld/rubiks.mp4)).
> >
> > This strategy is enabled by our two key design decisions: `(1)` verify & retry during data generation, and `(2)` distill diverse robot data down to diffusion policies. `(1)` ensures that robust recovery behavior is demonstrated during policy distillation and we show it leads to +47.5% in transport success (Table 2, revised PDF). However, simply retrying until success is not sufficient if retrying attempts are not diverse (e.g., identical to the previous attempt), which is addressed by `(2)`. Our policy’s learned robust strategy is the synergy resulting from both of these decisions.
> >
> > Please let us know if you have any further questions regarding the additional results!

---

### Official Review · Reviewer_bJic · 2023-07-23

**Confidence:** 3
**Originality:** Good
**Technical Quality:** Good
**Clarity Of Presentation:** Good
**Impact:** 2

**Recommendation:**

Weak Reject: I recommend rejecting the paper, but will not argue for my recommendation if the majority of other reviewers have a different opinion.

**Review:**

Fundamentally, the insight is that the data an agent can get via imitation is limited and expensive to collect but that sampling from existing components allows for testing boundaries and compositions that are still valid. This is nicely emphasized in the catapult case where the motion planner allows for atypical but successful placements.

See questions before for better understanding the effects of the composition of components

This is tied to the limitations of this work -- where do we already have robust components? is there a way to use the proposed verification loop to learn new skills?

**Quality Of The Limitations Section:**

Limitations are addressed clearly

**Questions For Rebuttal:**

My primary concern with is understanding the sources of and propagation of error -- which helps understand what would be necessary to extend the work:
1. Error from the motion profiles
2. Error from LM success detection?  How do we trust the verification? What's the process for encoding the scene? and is verification only possible because the work is done in simulation?
3. Error from incorrect Task Plans generations?
4. How do the accuracy of the examples stored align with human perception of success?
5. How expensive is the full process? (e.g. lines 267...) What percent of trials are failures (line 149...) and how computationally expensive is the trial process? --> Edit: This is partially addressed by Appendix 3.1
6. How many samples are being evaluated in Table 2?


**Robotics Focus:**

Sufficient demonstration on hardware

**Summary Of Paper:**

This work introduces an iterative procedure for data augmentation where task plans are generated, executed, and success verified.  The approach is simple but builds on several trained components to ensure the accuracy of the resulting data.  I am in agreement with the philosophical claims of collecting and learning alignments to motion.


Note: There are some rendering issues with figures that requires they be opened in adobe reader, other (Mac preview, Browser, etc) are not showing Fig 3 or lines in Fig 5. The figures seem to be very heavy so CPU jumped to 100% trying to render them.

**Summary Of Recommendation:**

Fundamentally, the idea is simple, appears to work well, but I am having trouble understanding the likelihood that other researchers would be able to leverage the approach or what the boundaries are of where it is most useful (e.g. Sim vs Real assumptions, MP quality, ...).  The single biggest gain seems to be related to distillation which is mentioned throughout but details are sparse, so it's difficult to understand what's unique versus retraining.

---

> ### Author Response · Authors · 2023-08-14
> **Rebuttal Follow-up**
>
> We want to thank you again for the great suggestions and clarifying questions. During the rebuttal, we have:
>  - Analyzed the sources and propagation of error for the longest horizon task, and provided the Table below (identical to Table 3 in the revised PDF).
>  - Added extra simulation experiments (Table 2 in revised PDF) as well as real world experiments clarify our contribution regarding policy distillation. Qualitative results on both domains are on our [anonymous website](https://scalingup-distillingdown.github.io/).
>  - Answered all your questions in our response below.
>
> We hope we have addressed all points raised in your review. Please don't hesitate to ask for more details.

---

### Author Response · Authors · 2023-08-11
**Response Summary, Revised PDF in each reviewer's thread**

We thank all the reviewers for their constructive feedback. We're glad the reviewers found our paper clear and well-written (t1nk, kQce), the approach simple (t1nk, bJic) yet novel (kQcE) and effective (bJic), and the benchmark diverse and challenging (t1nk).

Following the reviewer's suggestions, we've made the following important changes:
 - **Sim2Real**: Using domain randomization, we've deployed our distilled policy in the real world, without any real-world data.
 - **Sources and Propagation of Error (Failure Analysis)**: We've added language model planning and success detection accuracies, as well as execution success rates.
 - **Open-source Language Model**: We've included results using the open-source LLama2 models (7B, 13B).

We've made all changes in the paper highlighted in blue, and added qualitative results for both real world and (more) simulation on the supplementary website. We've attached the revised paper in every direct response to reviewers, since we weren't able to include files in this official comment.

---

### Decision · Program_Chairs · 2023-08-30

**Decision:**

Accept (Poster)

**Comment:**

The paper combines ideas to scale up the collection of vision+language+motor data with novel approaches to train language conditioned policies (based on Diffusion Policy). The general approach is highly relevant to bringing strongly generalizing LLM to robotic manipulation.

The reviewers appreciate the novel ideas. The rebuttal addressed many technical points raised by the reviewers. (The 2nd reviewer indicated potentially raising the score.) A limitation is the evaluation in simulation only. The authors mention 'the Sim2Real route as a good way for our method to be applied to the real world'.